# The Pulmonary Venous Return from Normal to Pathological—Clinical Correlations and Review of Literature

**DOI:** 10.3390/medicina57030293

**Published:** 2021-03-22

**Authors:** Cristina Claudia Tarniceriu, Loredana Liliana Hurjui, Daniela Maria Tanase, Alin Horatiu Nedelcu, Irina Gradinaru, Manuela Ursaru, Alexandra Stefan Rudeanu, Carmen Delianu, Ludmila Lozneanu

**Affiliations:** 1Department of Morpho-Functional Sciences I, Discipline of Anatomy, “Grigore T. Popa” University of Medicine and Pharmacy, Universității str 16, 700115 Iasi, Romania; claudia.tarniceriu@umfiasi.ro (C.C.T.); alin.nedelcu@umfiasi.ro (A.H.N.); 2Hematology Clinic, “Sf. Spiridon” County Clinical Emergency Hospital, 700111 Iasi, Romania; alexandrastefanumf@yahoo.com; 3Department of Morpho-Functional Sciences II, Discipline of Physiology, “Grigore T. Popa” University of Medicine and Pharmacy, 700115 Iasi, Romania; 4Central Clinical Laboratory—Hematology Department, “Sf. Spiridon” County Clinical Emergency Hospital, 700111 Iasi, Romania; carmendelianu@gmail.com; 5Department of Internal Medicine, “Grigore T. Popa” University of Medicine and Pharmacy, 700111 Iasi, Romania; tanasedm@gmail.com; 6Internal Medicine Clinic, “St. Spiridon” County Clinical Emergency Hospital, 700115 Iasi, Romania; 7Department of Implantology Removable Dentures Technology, “Grigore T. Popa” University of Medicine and Pharmacy, Universității str 16, 700115 Iasi, Romania; irina.gradinaru@umfiasi.ro; 8Department of Surgical Sciences, “Grigore T Popa” University of Medicine and Pharmacy, Universității str 16, 700115 Iasi, Romania; manuela.ursaru@umfiasi.ro; 9Department of Morpho-Functional Sciences I, Discipline of Histology, “Grigore T. Popa” University of Medicine and Pharmacy, Universității str 16, 700115 Iasi, Romania; ludmila.lozneanu@umfiasi.ro; 10Department of Pathology, “Sf. Spiridon” Emergency County Hospital, 700111 Iasi, Romania

**Keywords:** pulmonary veins, anatomical variants, anomalous pulmonary venous return

## Abstract

Pulmonary veins carry oxygenated blood from lungs to the left atrium of the heart. The anatomy of the pulmonary veins is variable with some anatomic variants. In clinical practice the difference between the normal anatomy of pulmonary veins with its variants and abnormal anatomy is very important for clinicians. Variants of pulmonary veins may occur in number, diameter and normal venous return. We present a case report and a review of the literature with the pulmonary venous return that deviates from the usual anatomical configuration and ranges from normal variant drainage to anomalous pulmonary—systemic communication. Initially, it was considered as an anatomical variant of the pulmonary venous return associated with the persistence of the left superior vena cava. Upon detailed exploration it was established that it was an anomaly of the pulmonary venous return which led in time to the installation of its complications. Diagnosis can be difficult, sometimes missed, or only made late in adulthood when complications were installed. Knowledge of variant anatomy and anomalous pulmonary venous return play a crucial role in the diagnostically challenging patient.

## 1. Introduction

Pulmonary veins are four in number and carry oxygenated blood from lungs to the left atrium of the heart [1]. The anatomy of the pulmonary veins is variable with some anatomic variants. The difference between the normal anatomy of pulmonary veins with its variants and anomalous anatomy is important in clinical practice and sometimes can be challenging for clinicians. Variants of pulmonary veins may occur in number, diameter, and normal venous return [1]. The pulmonary venous return can deviate from the usual anatomical configuration and ranges from normal variant drainage to anomalous pulmonary-systemic communications [2]. This means that the pulmonary veins are directly connected to the systemic venous circulation by maintaining a connection to the splanchnic circulation. The anomalous pulmonary venous return is divided into partial and total anomalous pulmonary venous return (TAPVR). Total anomalous pulmonary venous return (TAPVR) is a rare and critical congenital vascular anomaly, makes up 1.5% (1–3%) of congenital cardiac disease and the prevalence rate reaches up to 0.8/10.000 of live births [3]. The majority of cases have insufficient data to provide a hereditary predisposition. Patients with TAPVR have been reported occasionally with Holt–Oram syndrome, Ivemark syndrome, or Noonan syndrome [4,5]. In children, the order of frequency of types of TAPVR is supracardiac, 45%; infracardiac, 25%; intracardiac, 25%; and mixed, 5% [6]. The incidence in adults is unknown, the literature reports only the cases of late diagnosis. Compared to TAPVR, persistence of the left superior vena cava is the most common vascular abnormality found in the thorax, being found in approximately 0.3–0.5% of the general population [7]. The majority of patients suffering from TAPVR begin to experience symptoms in the first year of life and about 80% of suffering patients will die before their first birthday if left untreated [8,9]. The risk factors that influence a patient’s survival prognosis lay in the diagnosis of an atrial septal defect, an unobstructed abnormal pulmonary venous drainage, as well as a close to normal vascular pulmonary resistance [9]. We want to highlight the challenging diagnosis and anatomic clinical correlations of a patient that has a deviation from normal variant of pulmonary venous return, complicated with secondary polycythemia, secondary hypertension, and cerebrovascular stroke.

## 2. Case Report

A 27 year old male was admitted to hospital for a neurologic symptomatology including, vertigo, headache, blurred vision, right hemiparesis, and, upon clinical examination, he presented lower left back pain, mild perioral (central) and peripheric cyanosis, facial erythema, and grade 3 clubbed fingers. The patient was first admitted to hospital based on the diagnosis of a subacute ischemic stroke in the neurological sector of a university hospital, confirmed by a native cranial CAT scan (computed axial tomography), which revealed a hypodense non homogenous area and, postcontrast, a hyperdense image was seen, localized at the internal capsule and lenticular nucleus alongside the left corona radiata. Such a location provokes a discreet pushing effect on the lateral ventricle, measuring 58 × 21 × 42 mm³. Furthermore, the patient’s high cell blood count and high blood pressure (210/100 mmHg) posed a question for a differential diagnosis, that will be discussed later. Since the patient had polycythemia, it was thought that he could have a myeloproliferative disease and was thus transferred to the Haematology sector of the county emergency hospital. Furthermore, the pulse oximetry saturation was 80%. The patient’s heart sounds were well defined and a split S2 was identified. The haemoglobin value was 22.2 g/dL (normal value: 14–16 g/dL) and the haematocrit value was 64.2% (normal value: 35–45%). Upon further paraclinical investigations, the electrocardiogram identified prominent P waves, right ventricular hypertrophy, right bundle branch block and right axis deviation. After a session of phlebotomy, the patient’s haematocrit (57%) and haemoglobin (19.2 g/dL) values decreased. The patient maintained high blood pressure, and thus a secondary cause of hypertension was brought into discussion and led to further testing. The chest X-ray showed a smaller aortic arch with vascular prominence and an enlarged heart. Transthoracic echocardiography showed, on the parasternal long axis, a dilated coronary sinus and a right ventricle. The free wall of the right ventricle is hypertrophic, indicating higher pressure in the right heart chambers. On the parasternal short axis, which identifies the right and left ventricle, the diameters of the right ventricle is greater than that of the left ventricle, indicating greater pressure in the right heart chambers. The aetiology of this phenomenon was thought to originate from a possible disease that evolves with hypoxia.

This led to further investigations and a thoracic CAT scan was performed. The thoracic CAT scan showed a vertical collecting canal (vein) located in the left side of the mediastinum that was considered to be a persistent left superior vena cava associated with an anatomical variation of the pulmonary venous return (Figure 1A,B). Upon detailed exploration and reconstruction, images revealed that the pulmonary venous drainage takes place above the heart, and all four pulmonary veins are ending into a common collecting canal and then this trunk opens into the left brachiocephalic vein that drains into the superior vena cava (Figure 2). It was not noted a connection (opening) between the collecting canal and the left atrium. Upon a closer investigation and interpretation of the thoracic CAT scan images, the patient was seen to have an atrial septal defect, and therefore oxygen rich blood was sparse due to a right-to-left shunt and drains blood into the left atrium (Figure 3).

Furthermore, cardiac catheterization concurred with anterior findings and a total anomalous pulmonary venous return was noted and the blood drained into a collecting canal of approximately 41 mm through a permanent sinus of a vertical vein that drained into the left brachiocephalic vein (superior vena cava) and into the right atrium. Pulmonary hypertension was observed and measured to be 86/46 mmHg, aortic pressure measured 200/105 mmHg and the left–right shunt oximetry quantification was 83% oxygen saturation in the aorta and 91% oxygen saturation in the pulmonary artery. It is to be noted that the patient was diagnosed with an atrial septal defect that would prove life-saving in contingency with the anomalous pulmonary venous return.

## 3. Discussion

The pulmonary veins originate from capillary networks in the alveolar walls, are devoid of valves and return oxygenated blood to the heart. The intrapulmonary part of pulmonary veins is not located near the bronchi, follows the intersegmental septa and can be differentiated from the segmental pulmonary arteries [1]. Pulmonary veins exit from the lung through pulmonary hilum where they are located anterior and inferior to the pulmonary arteries in both sides (superior pulmonary vein is the most anterior structure and inferior pulmonary vein is the most inferior structure of the pulmonary hilum) [10]. Normal anatomy consists of four pulmonary veins in number, two for each lung and opened individually into the left atrium. The right superior pulmonary vein drains blood from right superior and right middle lobes. The right inferior pulmonary vein drains blood from the right inferior lobe. The left superior pulmonary vein drains blood from the lingula and left superior lobe of the left lung, whilst the left inferior lobe is drained by the left inferior pulmonary vein. This usual anatomical arrangement is found in 60–70% of people [2]. In 57–82% of population, four distinct pulmonary vein ostia are seen in the posterior wall of the left atrium [11]. Two of these orifices are on the right, draining the right superior and inferior pulmonary veins and two are on the left, draining the left superior and inferior pulmonary veins. In the present case, four pulmonary veins were identified, two that drain the venous blood from the right lung and two that drain the venous blood from the left lung. However, the four openings of the pulmonary veins in the left atrium were not highlighted.

A large variation of normal variant pulmonary veins exists and is an incidental finding, without clinical impact. Typical anatomical variants of pulmonary venous return take into account the presence of supernumerary veins or the fusion of some of them with an impact on the number of visible orifices in the left atrium. The results of imaging studies published in the literature have shown that most anatomical variants of pulmonary venous return consider right pulmonary venous return. So, Marom et al. [12] showed that right-sided venous drainage was more variable than left-sided venous drainage and one-quarter of patients had more than two venous ostia on the right side. Usually, when anatomical variants occur, the right pulmonary venous return is seen to be more complex and has one or more accessory veins [13]. An accessory vein has its own proper atrio-pulmonary venous junction separated from the superior and inferior pulmonary veins and is typically smaller [13]. Ghaye et al. [14] draw attention to the importance of making a differential diagnosis with an ostial branch that has not its own proper atrio-pulmonary venous junction. Often, it is described the accessory vein draining the middle lobe of right lung, which is seen in up to 26% of patients [12]. The left pulmonary venous return tends to be more simplified, often having the veins that converge into a short or long common trunk which drains into the left atrium [1,2,10,13].

Given the fact that in our case the four pulmonary veins were highlighted at the initial exploration, firstly it was considered that it was an anatomical variant of pulmonary venous return. It was considered that both left pulmonary veins join in a common trunk and two right pulmonary veins remain separate. The vertical collecting canal in the mediastinum was considered to be the persistence of the left superior vena cava. On a detailed re-examination and corroborated with the images of reconstruction and cardiac catheterization it was found that all four pulmonary veins do not open in the left atrium but they collect into a vertical canal (vein) that later opens into the left brachiocephalic vein, establishing diagnosis of a total anomalous pulmonary return. For a certain diagnosis it is very important to know these anatomical variants, as well as to establish the differential diagnosis with vascular development abnormalities.

Understanding of the embryology is important in appreciating the patterns of anomalous pulmonary return and its clinical impact. Three main aspects intervene in the embryological development of the thoracic and pulmonary venous systems: (a) the development of the systemic veins and the establishment of the connection with the venous splanchnic plexus; (b) the development of the pulmonary veins and the establishment of the connection with the intrapulmonary venous plexus; and (c) the involution of the systemic veins, with the interruption of the connection with the splanchnic plexus and the separation of two venous systems. The veins of the embryonic thorax consist of two pair large veins: superior cardinal veins and inferior cardinal veins. Superior cardinal vein drains the venous systemic blood from the cranial part of embryo and inferior cardinal vein from its caudal part. Superior and inferior cardinal veins join on each side to form right and left common cardinal veins before entering into the heart [15].

In humans, at day 26 of gestation, the right and left lungs appear as lung buds from the ventral wall of the primitive foregut [16]. The first pulmonary vessels are formed as a plexus in the mesenchyme surrounding the lung buds by vasculogenesis—splanchnic plexus [16]. As pulmonary development progresses, part of the splanchnic plexus forms the pulmonary plexus. The splanchnic plexus communicates with the systemic cardinal and umbilical—vitelline veins, thus forming the initial route of pulmonary venous drainage. Later, systemic veins involute, separating the two venous systems. Another important stage in the embryonic development is the development of the pulmonary veins and the establishment of the connection with the intrapulmonary venous plexus. There is no consensus about whether the pulmonary vein as a branch from the left atrium obtains a connection to the lung plexus or the pulmonary vein forms as a solitary vessel in the dorsal mesocardium [17] and is only secondarily incorporated into the atrium [18]. When the septum primum just begins to form, there is a single pulmonary vein which opens into the left atrium. The single pulmonary vein divides into two branches: right and left branch. Each of these branches bifurcates to drain the corresponding lung buds. Gradually, the parts of pulmonary veins nearest to the left atrium are absorbed into the atrium and four pulmonary veins, and two from each side come to open into left atrium.

In our case the diagnosis is a total anomalous pulmonary venous return which is an embryological defect of the primordial pulmonary veins. Total anomalous pulmonary venous connection results from failure to establish a normal connection between the pulmonary venous plexus and the common pulmonary vein and the connections with splanchnic venous system are not regressed (Figure 3). There is the absence of direct connection between any pulmonary veins and the left atrium. All the pulmonary veins drain to the right atrium by varied route: the vertical collecting canal, the left brachiocephalic vein, the superior vena cava that opens into right atrium (Figure 3).

The TAPVR classification most readily used is based on a definitive criterion linked to the anatomical venous connections, proposed by Darling [19]. There are four types of anomalous pulmonary venous return. The first type (type I = supracardiac) of TAPVR assures pulmonary venous drainage via a vertical vein that flows directly into the superior vena cava. This is the most commonly occurring type, affecting up to 50% of TAPVR cases [9]. In type II (cardiac), the venous drainage flows directly into the right atrium or the coronary sinus. In type III (infracardiac), the pulmonary venous return reaches the right atrium by way of the inferior vena cava. In type IV (mixed), the venous connections are situated both in the supracardiac, as well as the infracardiac position. The case presented is comprised of type I TAPVR characteristics (Figure 3). However, the patient’s age and method of diagnosis are uncommon. The congenital cardiac anomaly was not the primary reason for admission. It is common that patients at this age with TAPVR have signs of right ventricular overload [20]. It is important to note that the atrial septal defect was the key to this patient’s survival. It was also necessary to perform a differential diagnosis with persistence of the left superior vena cava, which is the most common vascular abnormality in the thorax [21] and with an anatomical variant of pulmonary venous return. Even though echocardiography is a sensitive and targeted method used in the diagnosis of TAPVR, in the adult patient this is not so. Thus, the golden standard in diagnosing an adult patient is a CAT scan or magnetic resonance imaging (MRI). These methods are more sensitive and can better target anomalies of the soft tissue. This is important especially in preoperative planning due to the exact details and it is able to provide information about the malformation [22]. The vascular development abnormalities were complicated with secondary polycythemia, secondary hypertension and cerebrovascular stroke. Secondary polycythaemia most often develops as a response to chronic hypoxemia, which triggers increased production of erythropoietin by the kidneys [23]. Increased red blood cell mass can elevate blood viscosity, which can impair blood flow, making individuals susceptible to vaso-occlusive events [24] An important clinical aspect is that the patient first presented neurological symptomatology, confirmed by a cranial CAT scan. The ischemic stroke can be produced by secondary polycythemia which decreases the cerebral blood flow by hyperviscosity and also by the secondary arterial hypertension. There is no clear incidence of TPAVR in adults, being reported only case presentations, which become unique by the age of diagnosis and the type of TPAVR. Our case is unique not only in the fact that it was diagnosed in adulthood, but especially in the way it is diagnosed. Compared to the TPAVR case presentations in the literature, the method of diagnosing of this vascular anomaly is unusual. The patient was diagnosed due to the installation of complications caused by the vascular anomaly of the pulmonary return. Initially, the patient was sent to the hematology clinic with the suspicion of chronic myeloproliferative syndrome, given by the high values of hematocrit. The key of diagnosis, was to exclude the chronic myeloproliferative syndrome diagnosis and to detect a cause for secondary polycythemia. In order to establish the final diagnosis, it was useful to know the anatomical variants of the pulmonary venous return and to exclude the diagnosis of persistence of the left superior vena cava. It is important to note that in our case, the atrial septal defect and absence of the pulmonary obstruction were the key to this patient’s survival. The main treatment is surgical repair of the vascular anomaly, the patient is being sent to a specialized center to establish the surgical technique. In this case, the operative risks are higher given by the complications already installed (pulmonary hypertension, secondary polycythemia, and ischemic stroke). These surgical risks could have been diminished if the diagnosis and surgical treatment had been performed in childhood.

## 4. Conclusions

Knowledge of the anatomy and normal development of the cardio-vascular system is essential for understanding the pathologies and for establishing anatomic clinical correlations. Highlighting of anomalous pulmonary venous return and variant anatomy is important and the radiologists play a crucial role in the diagnostically challenging patient. Furthermore, the secondary complication of this anomaly is also very significant from a clinical point of view.

## Figures and Tables

**Figure 1 medicina-57-00293-f001:**
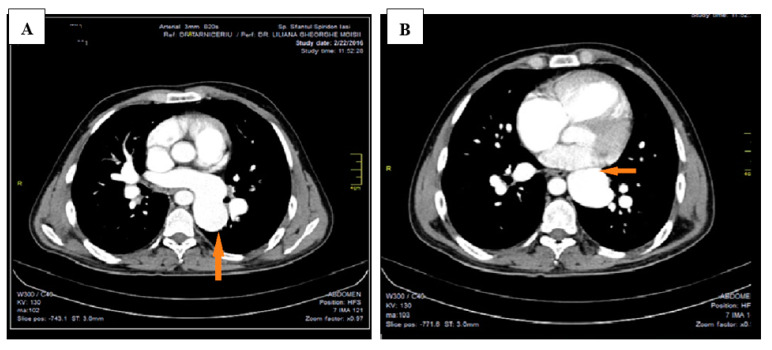
Thoracic CAT scan: (**A**)—a vertical collecting canal located in the left side of the mediastinum that opens into the left brachiocephalic vein. Arrow shows the presence of a vertical collecting canal.; (**B**)—not noted connection (opening) between the collecting canal and the left atrium. Arrow shows the relation of a vertical collecting canal with left atrium.

**Figure 2 medicina-57-00293-f002:**
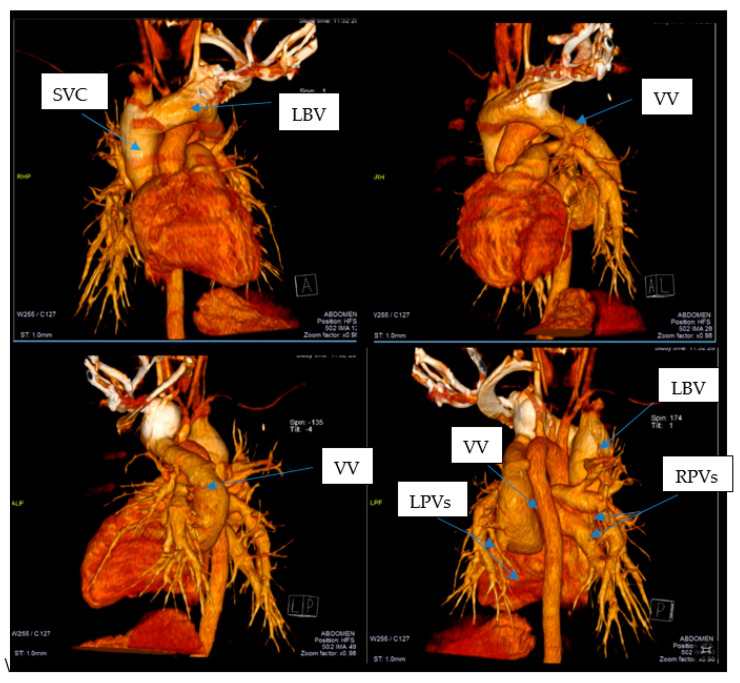
All four pulmonary veins are ending into a collecting canal that opens into the left brachiocephalic vein that drains into the superior vena cava. SVC—superior vena cava; LBV—left brachiocephalic vein; VV—vertical vein; LPVs—left pulmonary veins; RPVs—right pulmonary veins. Arrows shows the great vessels related to the heart.

**Figure 3 medicina-57-00293-f003:**
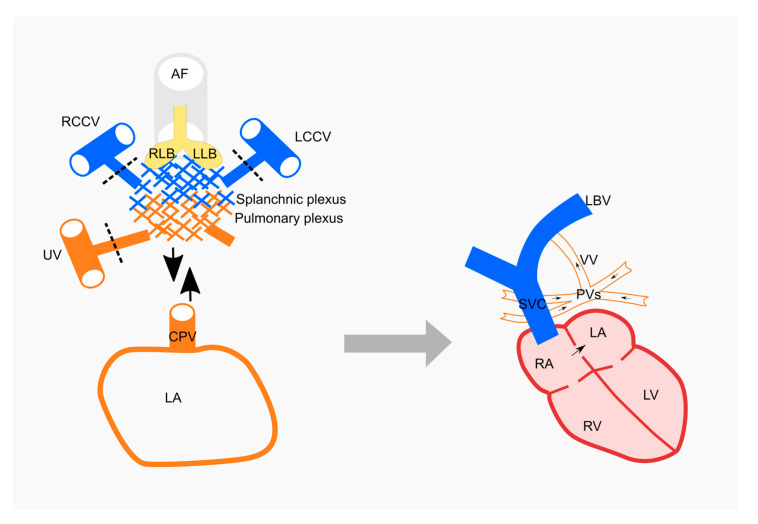
Type I = supracardiac of total anomalous pulmonary venous return (TAPVR) results from failure to establish a normal connection between the pulmonary venous plexus and the common pulmonary vein and the connections with splanchnic venous system is not regressed. All the pulmonary veins drain to the right atrium by varied route. AF—anterior foregut; RLB—right lung bud; LLB—left lung bud; RCCV—right common cardinal vein; LCCV—left common cardinal vein; UV—umbilical vein; CPV—common pulmonary vein; LA—left atrium; LVB—left brachiocephalic vein; VV—vertical vein; PVs—pulmonary veins; SVC—superior vena cava; RA—right atrium; RV—right ventricle; LV—left ventricle.

## Data Availability

The data presented in this case report are available on request from the corresponding author.

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
