# Peer review of "The Pulmonary Venous Return from Normal to Pathological—Clinical Correlations and Review of Literature"

_medicina, 2021, doi:10.3390/medicina57030293_

Round 1
Reviewer 1 Report
This is a case report. However, there is no novel finding in this patient. TAPVR is not so rare disease. The authors just described a generalization of TAPVR in the most part of this report. Although the patient’s age and method of diagnosis were not common, it does not support publication of this report.
Author Response
Review form 1
This is a case report. However, there is no novel finding in this patient. TAPVR is not so rare disease. The authors just described a generalization of TAPVR in the most part of this report. Although the patient’s age and method of diagnosis were not common, it does not support publication of this report.
Response:
Total anomalous pulmonary venous return (TAPVR) is a rare and critical congenital vascular anomaly, makes up 1.5% (1-3%) of congenital cardiac disease and the prevalence rate reaches up to 0.8/10.000 of live births. The majority of cases have insufficient data to provide a hereditary predisposition. Patients with TAPVR have been reported occasionally with Holt-Oram syndrome, Ivemark syndrome or Noonan syndrome. In children, the order of frequency of types of TAPVR is supracardiac, 45%; infracardiac, 25%; intracardiac, 25%; and mixed, 5% The incidence in adults is unknown, the literature reports only the cases of late diagnosis. Compared to TAPVR, persistence of the left superior vena cava is the most common vascular abnormality found in the thorax, being found in approximately 0.3-0.5% of the general population.
There is no clear incidence of TPAVR in adults, being reported only case presentations, which become unique by the age of diagnosis and the type of TPAVR. Our case is unique not only in the fact that it was diagnosed in adulthood, but especially in the way it is diagnosed. Compared to the TPAVR case presentations in the literature, the method of diagnosing of this vascular anomaly is unusual. The patient was diagnosed due to the installation of complications caused by the vascular anomaly of the pulmonary return. Initially, the patient was sent to the hematology clinic with the suspicion of chronic myeloproliferative syndrome, given by the high values of hematocrit. The key of diagnosis, was to exclude the chronic myeloproliferative syndrome diagnosis and to detect a cause for secondary polycythemia. In order to establish the final diagnosis, it was useful to know the anatomical variants of the pulmonary venous return and to exclude the diagnosis of persistence of the left superior vena cava. It is important to note that in our case, the atrial septal defect and absence of the pulmonary obstruction were the key to this patient’s survival.
Reviewer 2 Report
Thank you for giving me the opportunity to review the manuscript medicina-1095204, with the title “The pulmonary venous return from normal to pathological-clinical correlations and review of literature”. This is a very interesting case of congenital pulmonary vascular abnormality. The etiology, clinical presentation and image findings have been well reviewed by the author. I think it will provide useful information and knowledge to readers.
@@Some questions need to be clarified as following
- Introduction: Could you provide some data regarding the incidence or prevalence of “the anomalous pulmonary venous return” or “TAPVR”? The epidemiology data will provide readers how rare or unusual of cases you want to present in the article. Therefore, add on some incidence or prevalence data in the part of introduction as you mentioned in discussion is recommended.
2. Case reports: Figure 3 is very good picture for reader to understand the complicated anatomic structure in this case.
3. Discussion: 1. Could you provide some rationale and how unique in this case for a 27 y/o patient diagnosed as “TAPVR”? As you mentioned in the introduction, “The majority of patients suffering from TAPVR begin to experience symptoms in the first year of life and about 80% 56 of suffering patients will die before their first birthday if left untreated…” 2. The content of discussion could more focused and condensed to let readers catch the important points from this articles more easily .
Author Response
"Please see the attachment."

Reviewer 3 Report
I read the article with interest. Overall, it is well written, with an interesting case. I congratulate the authors for working out the variation in the anatomy of this case which could be challenging. It is important to understand the variants of the pulmonary venous return including in this case it helped the clinicians to work out potential reason of the polycythaemia and stroke.
I have a few questions and comments which I hope will help:
- Could the authors comment a bit more the epidemiology including the incidence/prevalence of these variants? Are they associated with any conditions/syndromes?
- What are the potential treatment including any surgical intervention, particularly in this case? Would there be any treatment if this variant has been detected much earlier?
- Please comment on what this case adds into the literature? Any novel findings?
Author Response
"Please see the attachment."

Round 2
Reviewer 1 Report
I have no further comments.
This manuscript is a resubmission of an earlier submission. The following is a list of the peer review reports and author responses from that submission.